# Track Irregularity Identification Method of High-Speed Railway Based on CNN-Bi-LSTM

**DOI:** 10.3390/s24092861

**Published:** 2024-04-30

**Authors:** Jinsong Yang, Jinzhao Liu, Jianfeng Guo, Kai Tao

**Affiliations:** Infrastructure Inspection Research Institute, China Academy of Railway Sciences, Beijing 100081, China; yangjinsong1994@163.com (J.Y.); guojf@rails.cn (J.G.); clarktao@rails.cn (K.T.)

**Keywords:** track irregularity, body vibration acceleration, Bi-LSTM, high-speed railway

## Abstract

Track smoothness has become an important factor in the safe operation of high-speed trains. In order to ensure the safety of high-speed operations, studies on track smoothness detection methods are constantly improving. This paper presents a track irregularity identification method based on CNN-Bi-LSTM and predicts track irregularity through car body acceleration detection, which is easy to collect and can be obtained by passenger trains, so the model proposed in this paper provides an idea for the development of track irregularity identification method based on conventional vehicles. The first step is construction of the data set required for model training. The model input is the car body acceleration detection sequence, and the output is the irregularity sequence of the same length. The fluctuation trend of the irregularity data is extracted by the HP filtering (Hodrick Prescott Filter) algorithm as the prediction target. The second is a prediction model based on the CNN-Bi-LSTM network, extracting features from the car body acceleration data and realizing the point-by-point prediction of irregularities. Meanwhile, this paper proposes an exponential weighted mean square error with priority inner fitting (EIF-MSE) as the loss function, improving the accuracy of big value data prediction, and reducing the risk of false alarms. In conclusion, the model is verified based on the simulation data and the real data measured by the high-speed railway comprehensive inspection train.

## 1. Introduction

The smoothness of the track directly determines the safety of the track system and passenger riding comfort [1]. At present, many countries have carried out the reform of track maintenance systems. In order to improve maintenance efficiency and save maintenance costs, the application of fault prediction and health management methods in track maintenance systems is gradually increasing. Replacing the traditional periodic maintenance mode with condition-based maintenance has become the development trend of track maintenance systems. The core of implementing condition-based maintenance is to use a great amount of detection data to scientifically and reasonably judge the smooth state of a track. Therefore, there are higher requirements for track detection frequency and accuracy.

With the continuous growth of high-speed railway traffic volume, in order to ensure the safety of railway operations, inspection frequency needs to be increased accordingly. Therefore, more convenient track smoothness detection methods have become research key topics.

At present, research on track irregularities mainly includes two aspects: one is the maintenance technology for track irregularities, and the other is detection technology and recognition algorithms. In terms of maintenance technology, Zhang et al. [2] proposed a curve radius optimization method to solve the problem of vehicle vertical acceleration exceeding the limit on vertical curves. In terms of identifying track irregularities, they can be directly detected through detection equipment, and track smoothness can be judged through statistical analysis of detection data [3]. It can also be indirectly evaluated through detection data such as vehicle dynamic response and wheel vibration noise [4]. There are mainly three kinds of track irregularity detection methods. The first is based on professional track detection vehicles, the second is the lightweight track survey trolley, and the third is the track detection system carried on in-service railway vehicles.

The professional track detection vehicle technology is relatively mature. It is generally equipped with a complex track detection system, which can measure the track geometry from different dimensions. For example, the HSR-350x track irregularity detection system in South Korea is composed of laser, camera, and inertial tools, and its maximum measurement speed is 320 km/h [5]. Tsubokawa et al. [6] proposed an inertial mid-chord offset method to detect track irregularity. This method combines the characteristics of optical and inertial methods and reduces the integration error caused by inertial measurement.

Although the detection method based on a professional detection vehicle has high detection efficiency, the detection accuracy and positioning accuracy of a detection vehicle will not meet the use requirements in the case of construction or precise track adjustment. Therefore, the lightweight track survey trolley is also an important detection tool for track irregularity. At present, the main principle of the track survey trolley is to combine the inertial navigation system with the total station or global positioning system to realize the high-precision measurement of track geometry [7]. Chen et al. [8] designed a track geometry measuring trolley system by integrating an inertial navigation system (INS) with geodetic instruments, which effectively measures track irregularity. Zhu et al. [9] proposed an attitude variation method based on double difference GNSS (DGNSS) and INS integration for railway track irregularity detection. 

The first two kinds of detection equipment are equipped with a large number of optical and inertial sensors, which have high measurement accuracy. However, such inspections are expensive and can only be performed periodically, which means that track failures can occur between inspections [10]. Therefore, how to realize the prediction of track irregularity by carrying simple inertial sensors on active trains has become a research hotspot in recent years [11].

Xiaozhou et al. demonstrated the correlation between track irregularity and vehicle acceleration through fractal analysis and pointed out that the correlation coefficient exceeds 0.7 when the wavelength is greater than 30 m. The methods of predicting track irregularity through vehicle dynamic response can be divided into methods based on the vehicle dynamics model and methods based on the filter model [12]. Track irregularity prediction can be regarded as an inverse dynamic analysis problem, in which track irregularity is an unknown input to be identified, and track irregularity can be identified through axle box or vehicle body acceleration measurement [13]. Czop et al. [14] proposed a detection method of track irregularity based on axle box acceleration measurement during vehicle operation according to the vehicle dynamics model. Compared with the acceleration of the axle box, the detection of the acceleration of the vehicle body is simpler and can be made into a portable device. Tsunashima et al. [15] used the Kalman filter (KF) to predict the track geometry irregularity of the Shinkansen in Japan according to the acceleration of the vehicle body. Odashima [16] demonstrated the possibility of estimating the irregularity of a conventional railway track using only the acceleration of the car body and used a Kalman filter for inversion track irregularities. However, the current prediction methods often have better prediction effects under certain fixed line conditions [17] and speed ranges, and the prediction accuracy will decrease with the increment of the prediction distance. Therefore, it is of great significance to find a track irregularity prediction method that has a wide range of applications and can achieve long-distance prediction.

In recent years, as an artificial intelligence technology involving deep network structure, deep learning provides a new idea for processing time series signals such as vehicle acceleration [18].

The recurrent neural network (RNN) proposed by Hopfield [19] in 1982 is a deep neural network that can consider the time correlation in time series. RNN network has no limit on the length of time series and is widely used in natural language processing. However, there are problems of gradient explosion and gradient disappearance in the training process of RNN. In order to overcome these problems, Hochreiter [20] proposed long short-term memory networks (LSTMs) in 1997. At present, LSTMs are widely used by researchers in the fields of natural language translation [21,22,23], speech recognition [24], finance [25], and signal processing [26]. LSTMs are also commonly used in the field of industrial equipment health management. Wanqing et al. [27] compared LSTMs with recursive neural networks and generalized Cauchy methods and summarized their respective characteristics. 

In order to improve the effect of feature extraction, the LSTM neural network is often combined with the convolutional neural network (CNN) [28]. Wu et al. [29] proposed a new framework structure according to the characteristics of financial time series and the task of price prediction, which combines the convolutional neural network and LSTM neural network to realize a more accurate prediction of stock price.

In order to make full use of the information before and after time series, bidirectional LSTM has attracted more attention in recent years. Zou et al. [30] proposed a method combining multi-scale weighted entropy morphological filtering (MWMF) signal processing with bidirectional long short-term memory neural networks (Bi-LSTMs). This method is applied to rolling bearing fault diagnosis. It is verified that the model has high classification accuracy. Xia et al. [31] proposed an integrated framework based on convolution multi-time window Bi-LSTM, which is used to accurately predict the maintenance rules of mechanical equipment when the length of condition monitoring data is highly inconsistent.

In the existing track irregularity research, there are relatively few researches that use depth learning methods to realize point-by-point prediction of track irregularity based on vehicle body acceleration. Previous studies have shown that there is a correlation between track irregularity and car body acceleration [32,33,34], which provides mechanism support for the identification of track smoothness based on car body acceleration data.

The smoothness of the track is the foundation for ensuring the safe operation of the high-speed train. Based on the characteristics of the correlation between track irregularity and car body acceleration, this paper constructs the track irregularity prediction model of high-speed railway based on Bi-LSTM and realizes the prediction of track irregularity value through the car body vertical acceleration detection data. Because car body acceleration detection is simple and does not need professional inspection vehicles, this method can realize the prediction of track smoothness by collecting the car body acceleration data of operating trains. Using this method to predict track irregularities can increase detection frequency at a lower cost and enable more timely detection of track irregularities. On the basis of constructing the prediction model based on the CNN-Bi-LSTM neural network, and by taking into consideration of the characteristics of track irregularity prediction, this paper proposes an exponential weighted mean square error with priority inner fitting (EIF-MSE) as loss function. This improves the prediction accuracy of high-risk values and reduces the possibility of false alarms caused by higher predictive values.

The structure of the rest of this paper is as follows: Section 2 introduces the methods of data preprocessing and data set construction. Section 3 introduces the process of the track irregularity prediction algorithm based on the CNN-Bi-LSTM model. Section 4 introduces the exponential weighted mean square error with priority inner fitting proposed in this paper. In Section 5, the model is verified by using the simulation data and the real data detected by the comprehensive inspection train.

## 2. Materials and Methods

### 2.1. Trend Extraction Based on HP Filtering Algorithm

During the operation of the high-speed comprehensive inspection vehicle equipped with the track geometry inspection system, it is inevitable to be disturbed by vehicle vibration and weather and temperature changes, which will affect the accuracy of the detection results. External sunlight reflection, sensor and data transmission errors, laser deviation from the normal detection point at the turnout, image interference, and other reasons will lead to high-frequency noise in track geometry irregularity detection data. When we predict the track irregularity, we pay more attention to the changing trend and overall amplitude of the irregularity. Therefore, when building the data set, we first use the HP filtering algorithm to filter the low irregularity data, extract the change trend, and filter out the high-frequency fluctuations.

HP filtering was proposed by Hodrick and Prescott in 1981 and is widely used in cost-effective analysis. This method can extract the fluctuation trend in the original data and reduce the influence of high-frequency interference [35].

HP filtering was proposed in 1981 to address trend analysis issues in the financial field. The HP filtering method can extract trend components from raw data and reduce the interference of high-frequency components. In this article, the HP filtering method is used to extract trend components from track irregularity detection data for model training and testing. The process of using the HP filtering algorithm to process track irregularity detection data is as follows:

The sequence of track irregularity detection data is represented as $y\left(t\right)=\left\{y_{1},y_{2}\cdots ,y_{m}\right\}$, if the trend component is represented as $q\left(t\right)=\left\{q_{1},q_{2}\cdots ,q_{m}\right\}$, and the fluctuation component is represented as $g\left(t\right)=\left\{g_{1},g_{2}\cdots ,g_{m}\right\}$, then $y\left(t\right)=q(t)+g(t)$.

In the HP filtering method, the loss function Z is set to:(1)$$Z=\sum_{i=1}^{m}{\left(y_{i}-q_{i}\right)}^{2}+\beta \sum_{i=2}^{m-2}{\left(q_{i}-2q_{i+1}+q_{i+2}\right)}^{2}$$
where *β* is a penalty factor that controls the degree of smoothness. According to Ref. [36], the determination of *β* mainly depends on the detection frequency of monitoring data. Based on the characteristics of irregularity detection data, *β* is selected as 30. The problem of solving the trend component *q*(*t*) is transformed into the problem of solving the minimum value of *Z*, which can be simplified as:(2)$$Z={\left\|y-q\right\|}^{2}+\beta {\left\|\nabla^{2}q\right\|}^{2}$$
where, if
(3)$$\nabla q={\left[\begin{array}{cccccc} -1 & 1 & \; & \; & \; & \;\\ \; & -1 & 1 & \; & \; & \;\\ \; & \; & \; & \cdots & \; & \;\\ \; & \; & \; & \; & -1 & 1 \end{array}\right]}_{\left(m-1\right)\times m}q,$$
then we can get:(4)$$\nabla^{2}q={\left[\begin{array}{cccccc} -1 & 1 & \; & \; & \; & \;\\ \; & -1 & 1 & \; & \; & \;\\ \; & \; & \; & \cdots & \; & \;\\ \; & \; & \; & \; & -1 & 1 \end{array}\right]}_{\left(m-1\right)\times m}\nabla q$$

If D represents $\nabla^{2}$, it can be abbreviated as $\nabla^{2}q=Dq$, and the loss function can be transformed into:(5)$$\begin{array}{cc} Z & ={\left\|y-q\right\|}^{2}+\lambda {\left\|Dq\right\|}^{2}\\ \; & =(y-q{)}^{T}(y-q)+\beta {\left(Dq\right)}^{T}\left(Dq\right)\\ \; & =q^{T}\left(I+\beta D^{T}D\right)q-2y^{T}q+y^{T}y \end{array}$$

By calculating the gradient, we can obtain:(6)$$y=\left(I+\beta D^{T}D\right)q$$
which can be solved as follows:(7)$$q={\left(I+\beta D^{T}D\right)}^{-1}y,$$

The fluctuation trend of the time series can be obtained by Equation (7).

The track irregularity trend data obtained by the HP filtering algorithm will be used for the following model training and prediction.

### 2.2. Data Set Construction Based on Moving Sliding Window

The irregularity data and acceleration data collected by the inspection train are sampled at equal intervals with 4 sampling points per meter. The acceleration detection data corresponds to the irregularity detection data one by one. In order to facilitate model training, in this paper, a moving window with a length of 500 sampling points is used to intercept the detection data, the acceleration sequence composed of 500 points is used as the model input, and the corresponding irregularity sequence is used as the target of the model, as shown in Figure 1. Thus, a data set for model training and testing is formed.

## 3. Track Irregularity Prediction Model Based on CNN-Bi-LSTM

In this paper, with CNN and Bi-LSTM, a fusion network model is proposed. The characteristics of deep learning are used to be able to extract features independently to learn the deep features contained in the acceleration sequence. Figure 2 shows the proposed CNN-Bi-LSTM network learning framework. The CNN-Bi-LSTM network parameter configuration is as shown in Table 1.

In this paper, one-dimensional convolution is used for the feature extraction of car body vertical acceleration data. A one-dimensional convolutional network consists of a convolutional layer and activation function. Since the acceleration of the car body has positive and negative points, the ‘tanh’ function is selected as the activation function. The convolution core size of the convolution layer is set to 10, the step size is 1, and the number of convolution cores is 100. In order to ensure that the acceleration data passing through the convolution layer can correspond to the track irregularity data one by one, the padding method is used to ensure that the data length before and after convolution is consistent.

Because the response of car body acceleration to track irregularity has a certain delay, when predicting track irregularity through car body acceleration data, we should consider not only the acceleration data before the current point, but also the information after the current point. Based on this feature, this paper uses the Bi-LSTM neural network to extract the features of the time dimension.

In the whole model, the convolution network is used to extract the deep features in the acceleration detection sequence. Then, the Bi-LSTM network is used to extract time series features. The features obtained by the CNN network are propagated in both positive and negative directions, so as to obtain time series features. After the Bi-LSTM network, each node obtains a 100 × 1 characteristic sequence. Finally, the characteristic sequence output by each node is input into the fully connected network for regression. After the three-layer fully connected neural network, the track irregularity predictive value corresponding to this point is obtained.

## 4. Exponential Weighted Mean Square Error with Priority Inner Fitting

In past regression prediction problems, mean square error is a common loss function. However, track irregularity prediction has its own characteristics, which are mainly reflected in two aspects. First, pay more attention to the accuracy of big value prediction. We call the value with big deviation from the standard value “big value”, which often represents high safety risk and high possibility of defects. Second, try to avoid the predicted value being higher than the real value. For the general mean square error, when the loss function takes a certain value, it may be caused by the predicted value being greater than or less than the real value. However, in contrast, when the predicted value is greater than the real value, it may cause false early warning, and in serious cases, it may suspend the train operation and affect the transportation efficiency. Therefore, we hope that when the predictive error meets the use requirements, the absolute value of the prediction should be less than the real value, that is, give priority to inside forecast bias.

These two characteristics are not only applicable to the prediction of track irregularity, but also applicable to other fault diagnosis, state prediction, and so on. Based on the demand characteristics of the above-mentioned two forecasts, this paper proposes an exponential weighted mean square error with priority inner fitting (EIF-MSE).

The traditional mean square error can be expressed as:(8)$$MSE=\frac{1}{n}\sum_{i=1}^{n}{\left(y_{i}-{\hat{y}}_{i}\right)}^{2},$$
where $y_{i}$ represents the *i*-th in the real value sequence $Y=\left[y_{1},y_{2},\cdots ,y_{n},\right]$ and ${\hat{y}}_{i}$ represents the *i*-th in the corresponding prediction sequence, $\hat{Y}=\left[{\hat{y}}_{1},{\hat{y}}_{2},\cdots ,{\hat{y}}_{n},\right]$.

The EIF-MSE proposed in this paper can be expressed as:(9)$$EIF\text{-}MSE=\frac{1}{n}\sum_{i=1}^{n}{\left(y_{i}-{\hat{y}}_{i}\right)}^{2}*e^{\lambda \cdot \left|y_{i}-\tilde{y}\right|}+\gamma \cdot relu\left(\left|{\hat{y}}_{i}\right|-\left|y_{i}\right|\right),$$
where $e^{\lambda \cdot \left|y_{i}-\tilde{y}\right|}$ is the risk coefficient, $\tilde{y}$ is the standard value of predicted quantity y, and the value is 0 for track irregularity.

The greater the $\left|y_{i}-\tilde{y}\right|$ value, the greater the deviation of the real value from the standard value, and the greater the potential risk and hazard. In order to ensure the accuracy of large-value predictions, the loss here should be greater.

$\gamma \cdot relu\left(\left|{\hat{y}}_{i}\right|-\left|y_{i}\right|\right)$ is called the inboard coefficient, where:$$relu\left(x\right)=\left\{\begin{array}{c} x\\ 0 \end{array}\right.\begin{array}{c} ,\\ , \end{array}\begin{array}{c} x>0\\ x\leq 0 \end{array}$$

When the absolute value $\left|{\hat{y}}_{i}\right|$ of the predicted value is less than the absolute value $\left|y_{i}\right|$ of the real value, the inner coefficient is 0, which is equivalent to not increasing the penalty for the loss function. When the absolute value $\left|{\hat{y}}_{i}\right|$ of the predicted value is greater than the absolute value $\left|y_{i}\right|$ of the real value, the inner coefficient increases with the increase of $\left|{\hat{y}}_{i}\right|$, and the loss function is punished to ensure that the predicted value is preferentially located inside the real value.

## 5. Model Validation

### 5.1. Simulation Model Verification

In order to verify the effect of the track irregularity prediction model constructed in this paper and EIF-MSE proposed in this paper, the simulation data are used to verify the model.

The vehicle-track coupling vibration model of a high-speed EMU is established by using ANSYS 19.0and SIMPACK 2018. Taking the line deformation in the 32 m simply supported beam section as the irregularity input, the simulation data of car body acceleration are obtained. Then, the data set is constructed by using the simulated car body acceleration data and track irregularity data, and MSE and EIF-MSE are used as loss function training models, respectively. The test set data is used to verify the prediction effect of the model, and the results are shown in Figure 3.

Comparison demonstrates that using the EIF-MSE loss function has a better prediction effect at peak points similar to points A and B. Through comparison, it is also shown that the EIF-MSE loss function proposed in this paper has better prediction effect at peak value points similar to points A and B.

In order to verify the effect of the EIF-MSE loss function proposed in this paper, two indexes of peak error and inner deviation are constructed, respectively. The peak error is defined as:(10)$$PE=\frac{1}{n}\sqrt{{\left(y_{m}-{\hat{y}}_{m}\right)}^{2}},$$
where *n* indicates that there are *n* peaks in the test data and $y_{m}$ indicates the *m*-th of them. The accuracy of the model for high value prediction can be evaluated through the peak error value.

Outside error is defined as the percentage of cases where the absolute value of the predicted value exceeds 20% of the absolute value of the real value, as in Formula (9). The percentage reflects the role of the inner coefficient in the EIF-MSE loss function.
(11)$$OE=\frac{num\left(\left(\left|{\hat{y}}_{m}\right|-\left|y_{m}\right|\right)>0.2\left|y_{m}\right|\right)}{n},$$

Bi LSTM, LSTM, and RNN are used for comparison and verification, respectively. Each algorithm uses MSE loss function and EIF-MSE loss function to train the model, and then uses PE and OE to verify the model. The results are shown in Table 2.

The data in the table demonstrates that using a combination of the Bi-LSTM network and the EIF_MSE loss function can get the best results. It is obvious that when EIF_MSE is used as the loss function, the peak error is less when compared with that using MSE.

### 5.2. Verification of Measured Data

In order to verify the prediction effect of the model under actual working conditions, the real track irregularity and car body acceleration data collected by the comprehensive inspection train of China’s high-speed railway (as shown in Figure 4) are used for verification. During the data collection process, the high-speed comprehensive detection train’s driving speed is 300 km/h, and the wavelength range of track irregularities that can be detected is 1.5~120 m. the track irregularity detection system and the car body acceleration detection system are deployed on the same carriage, ensuring the synchronization of data collection. The track irregularity detection system uses the inertial reference method to measure through gyroscopes and displacement sensors. The collection of car body acceleration is carried out using an acceleration sensor installed on the bottom plate of the vehicle, which is located on the same cross-section as the track irregularity detection system. The track irregularity and car body acceleration are both sampled at equal intervals, with a sampling interval of 0.25 m.

In order to construct the data set required for model training, the sliding window method introduced in Section 2.2 is used to truncate the acceleration detection data and irregularity data at the same time. See Figure 5. 

The track irregularity data is filtered according to the HP filtering algorithm introduced in Section 2.1, and the fluctuation trend of track irregularity data is extracted as the prediction target. The filtering effect of 50 m irregularity detection data is as shown in Figure 6.

It shows from the figure that the new column after HP Filtering better retains the fluctuation trend of the original sequence, and the small fluctuations need less attention and have been effectively filtered.

In order to verify the effect of the Bi-LSTM model on track irregularity prediction, the LSTM model is compared with the LSTM and RNN models by using the same data set and loss function. The features extracted by CNN are input into these three networks, respectively. Finally, the same fully connected neural network is used for regression prediction. The loss function in the training process of these three models is as shown in Figure 7. It demonstrates from the figure that the prediction model based on Bi LSTM has higher prediction accuracy.

The prediction effect of three different prediction models on a 125 m-long section is as shown in Figure 8. The red line is the real irregularity fluctuation of the section, and the blue line is the prediction of three different models. It also shows that the prediction effect based on the Bi LSTM model is obviously better than the other two models.

In order to verify the effect of EIF-MSE proposed in this paper, the effects of the risk coefficient and the inner coefficient on the model are verified, respectively.

In order to verify the effect of the inner coefficient, under other conditions unchanged, set γ to different values to train the model and test it. The results are shown as in Figure 9. It shows that at points A and B, when γ = 0.3, compared with γ = 0, the inner fitting effect of the model is better. Due to the effect of the inner coefficient, there are fewer cases where the predicted value is greater than the true value. It also shows in Table 3 that as the value of γ increases, the situation where the predicted value is greater than the real value gradually decreases.

Similarly, in order to verify the role of the risk coefficient, set λ to different values under other conditions unchanged. The results are shown in Figure 10. It shows that among the five big value points A, B, C, D, and E, when λ = 0.2, the fitting effect of the model is better than that of other cases. It can be seen that the addition of the risk coefficient enables the model to approach the true value more closely at big values. It also shows in Table 4 that the peak error of the model is the smallest when λ = 0.2.

From the above experimental results, it shows that the CNN-Bi-LSTM model constructed in this paper has better prediction results than the CNN-LSTM model and the CNN-RNN model. Through the comparison, it shows that using EIF-MSE as the loss function can better predict the big value compared with the traditional MSE error, and can reduce the possibility of false positives through the preferential inner fitting.

## 6. Conclusions

In order to find a more convenient detection method for track irregularity, this paper recommends a method based on car body acceleration. It constructs a prediction algorithm based on the CNN-Bi-LSTM model, takes the car body acceleration as the model input, extracts the features through the CNN network and the Bi-LSTM network, and finally obtains the point-by-point prediction results of track irregularity with a fully connected neural network. In order to ensure the prediction accuracy of big values and reduce the probability of false positives, this paper proposes an exponential weighted mean square error with priority inner fitting. Experiments show that, compared with the traditional MSE loss function, when using EIF-MSE loss function as the model loss function, more accurate prediction results can be obtained. The EIF-MSE loss function proposed in this paper can also be applied to other prediction occasions that pay more attention to the prediction accuracy of big values. 

## Figures and Tables

**Figure 1 sensors-24-02861-f001:**
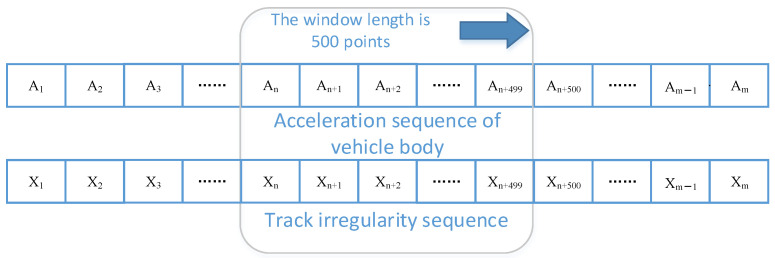
Data set construction method based on sliding window.

**Figure 2 sensors-24-02861-f002:**
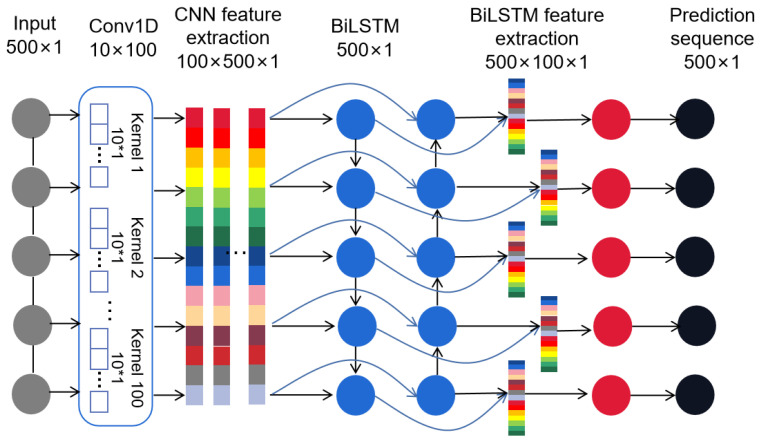
Model structure diagram.

**Figure 3 sensors-24-02861-f003:**
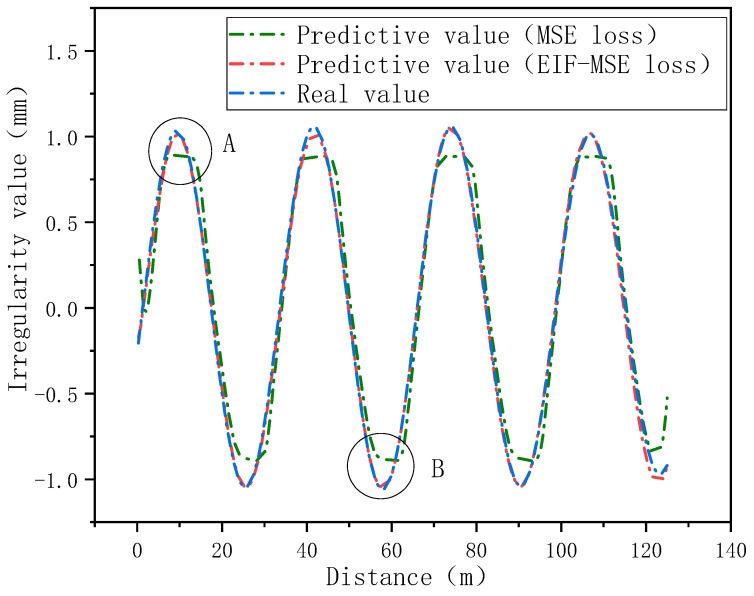
Model prediction effect based on simulation data (The positions A and B in the figure represent the peak areas of track irregularity).

**Figure 4 sensors-24-02861-f004:**
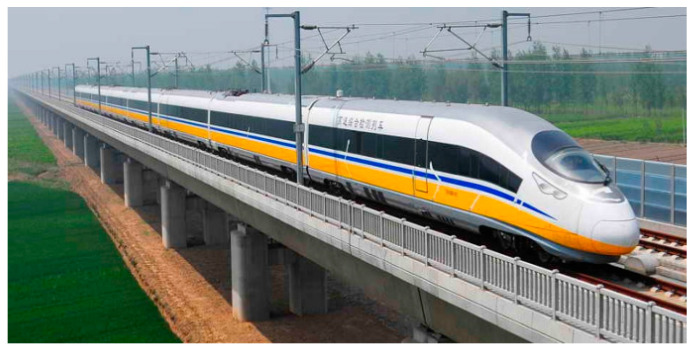
High-speed comprehensive inspection train.

**Figure 5 sensors-24-02861-f005:**
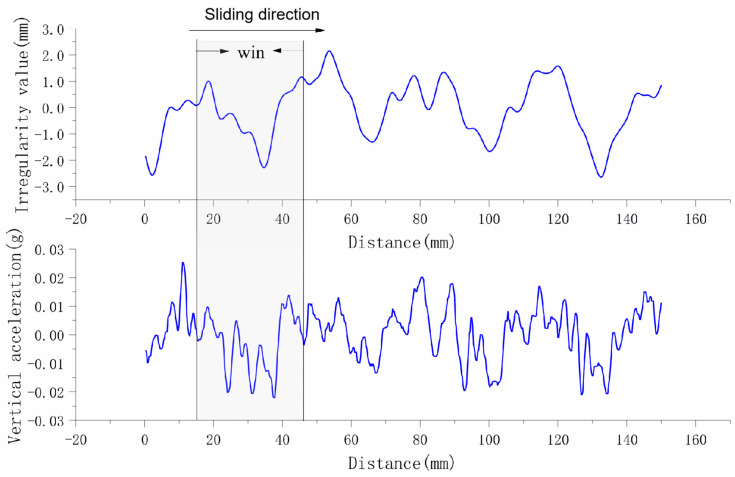
Data truncation method based on sliding window.

**Figure 6 sensors-24-02861-f006:**
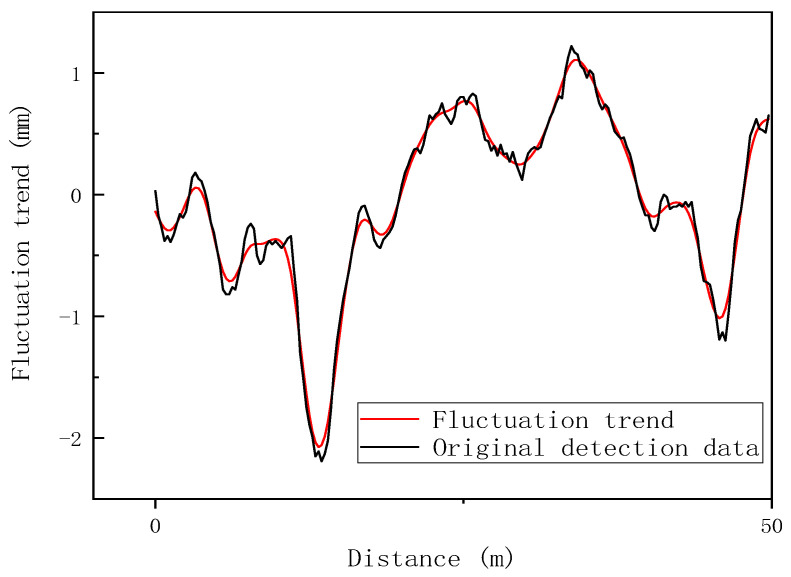
HP filtering effect.

**Figure 7 sensors-24-02861-f007:**
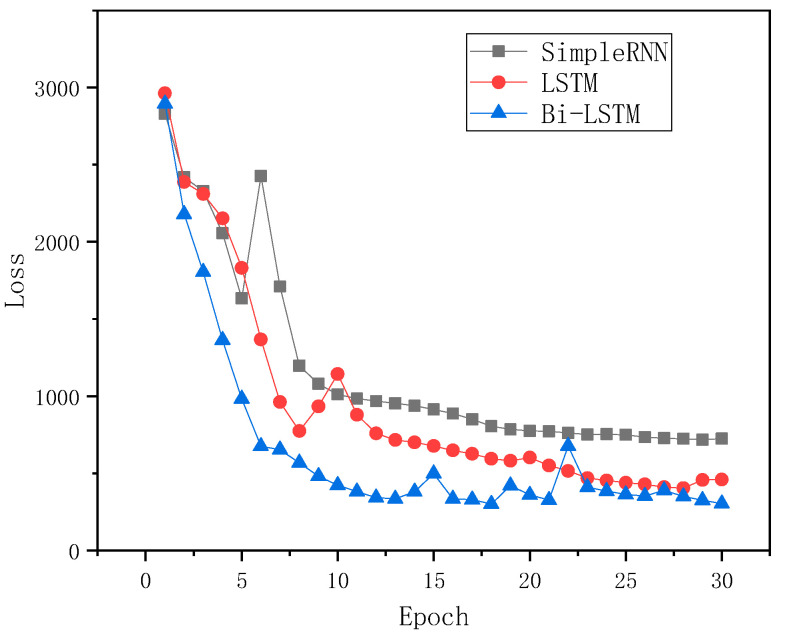
Comparison of training processes of different models.

**Figure 8 sensors-24-02861-f008:**
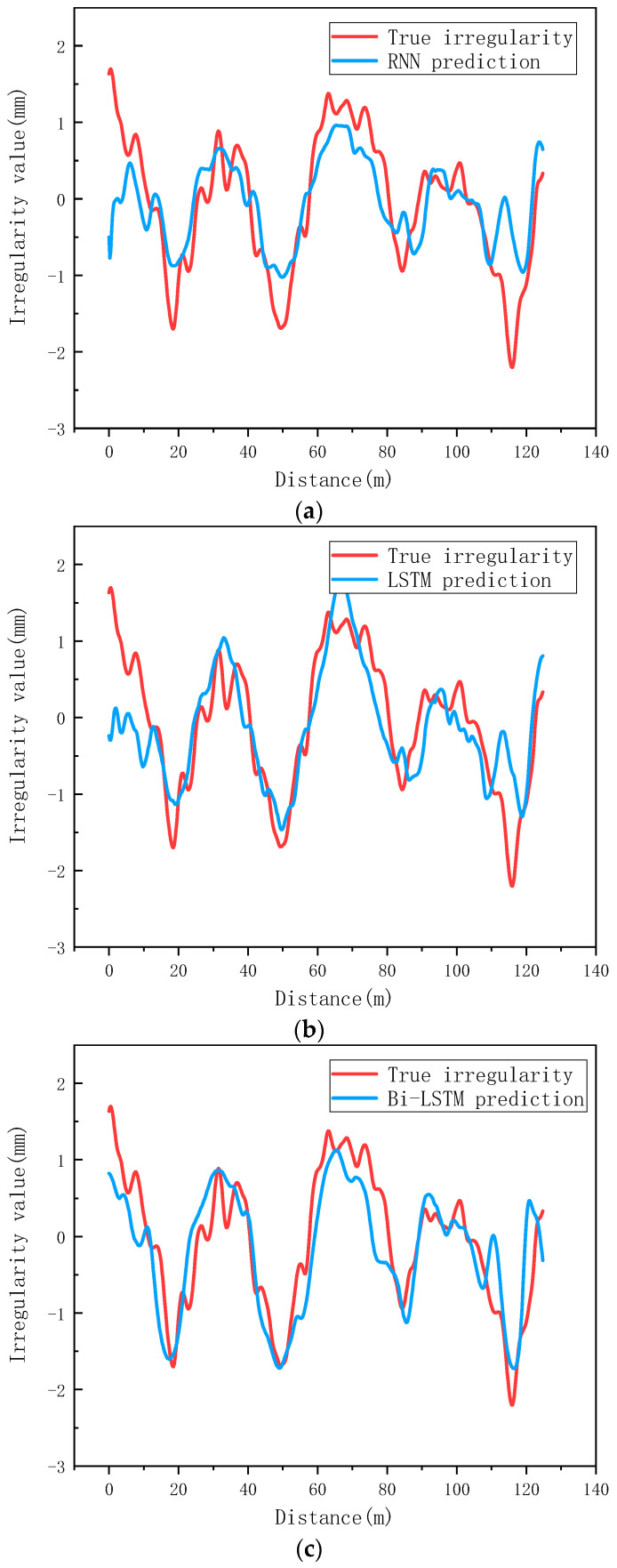
Comparison of prediction effects of different models: (**a**) Comparison of performance with RNN networks; (**b**) Comparison of performance with LSTM network; (**c**) Comparison of performance with Bi-LSTM network.

**Figure 9 sensors-24-02861-f009:**
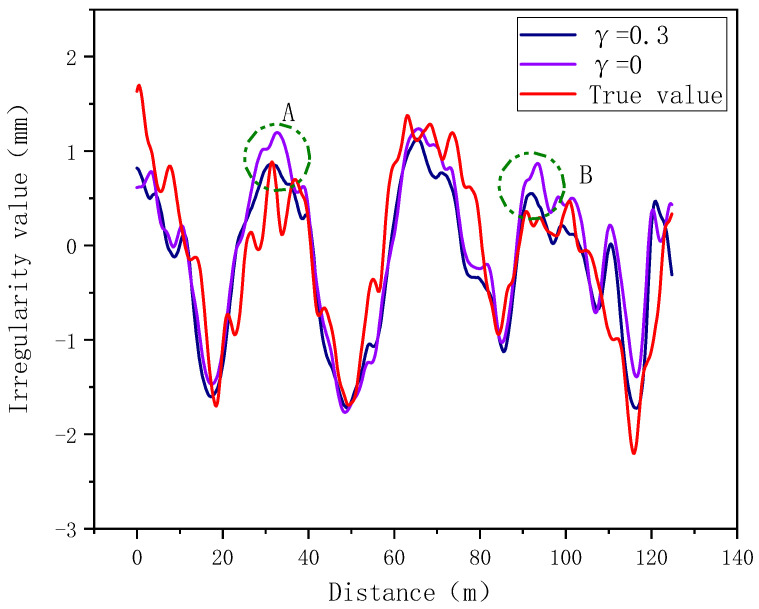
Internal verification effect verification (The positions A and B in the figure represent the peak areas of track irregularity).

**Figure 10 sensors-24-02861-f010:**
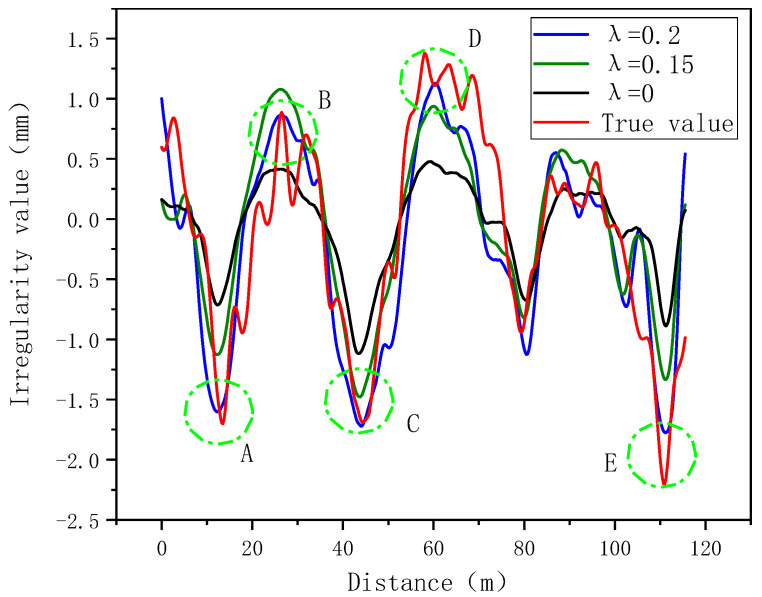
Risk coefficient effect verification (A~E represent five positions with big values).

**Table 1 sensors-24-02861-t001:** Parameters of models.

Model Level	Parameter
CNN	Conv1D (filters = 100, kernel_size = 10, activation = ‘tanh’)
Bi-LSTM	Bi-directional (LSTM(100, activation = ‘tanh’))
Dense	Dense (60, activation = ‘tanh’) Dense (30, activation = ‘linear’) Dense (1, activation = ‘linear’)

**Table 2 sensors-24-02861-t002:** Comparison of algorithm results.

Algorithm	RNN		LSTM		Bi-LSTM	
Loss function	MSE	EIF_MSE	MSE	EIF_MSE	MSE	EIF_MSE
PE	0.212	0.075	0.194	0.038	0.173	0.013
OE	16.4%	15%	15.6%	13.8	12.8%	12.2%

**Table 3 sensors-24-02861-t003:** Comparison of internal verification.

Parameter	γ = 0	γ = 0.3	γ = 1
OE	36%	30.6%	24.6%

**Table 4 sensors-24-02861-t004:** Comparison of risk verification.

Parameter	λ = 0	λ = 0.2	λ = 0.15
PE	0.475	0.171	0.276

## Data Availability

The data are available from the corresponding author on reasonable request.

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
