# Peer review of "Track Irregularity Identification Method of High-Speed Railway Based on CNN-Bi-LSTM"

_sensors, 2024, doi:10.3390/s24092861_

Round 1

Reviewer 1 Report

Comments and Suggestions for Authors

1. Some brief abbreviations should give an explanation table

2. the vibration peak value is the most important parameter for high speed train, authors should analyze the maximum error of the predicted peak. 

3. How to extract the vibration signal features by CNN features is not clear

4. as the prediction of long and short time memory,there are some up-date literature,suggest authors introduce them, for example: Long-range dependence and heavy tail characteristics for remaining useful life prediction in rolling bearing degradation.

5.When high-speed train is running, authors should explain that prediction vibration how to eliminate security problem?

Author Response

Dear reviewer,

Thank you very much for taking the time to review this manuscript. Please find the detailed responses below and the corresponding revisions highlighted.

Comments 1: Some brief abbreviations should give an explanation table

Response 1: Thank you for your suggestion. We have added a list of abbreviations at the end of the article

Comments 2: the vibration peak value is the most important parameter for high speed train, authors should analyze the maximum error of the predicted peak.

Response 2: Thank you for your suggestion. What you said is very accurate. In section 5.1 of the article, the model evaluation index PE was defined to evaluate the prediction error of the model on the peak value. In section 5.2, actual measurement data was tested, and the results are shown in Figure 10 and Table 4.

Comments 3:How to extract the vibration signal features by CNN features is not clear

Response 3:Thank you for your suggestion. In this article, 1D Conv was used to extract depth information from the original detection data and expand the dimensions. Figure 3 was modified and a CNN layer was added to make the model structure clearer

Comments 4:as the prediction of long and short time memory,there are some up-date literature,suggest authors introduce them, for example: Long-range dependence and heavy tail characteristics for remaining useful life prediction in rolling bearing degradation.

Response 4:Thank you for your suggestion. The literature you mentioned has important reference value for this article and has been added to the first section

Comments 5:When high-speed train is running, authors should explain that prediction vibration how to eliminate security problem?

Response 5:Thank you for the reviewer's suggestions. The smoothness of the track is the foundation for ensuring the safe operation of the high-speed train. This article predicts track irregularities through car body acceleration, which can increase detection frequency at a lower cost and detect track irregularities more promptly. According to your suggestion, relevant content has been added in the first section.

Reviewer 2 Report

Comments and Suggestions for Authors

The manuscript presents a method for track irregularity identification of high-speed railway based on CNN-Bi-LSTM, both simulation and the real data measured by the high-speed railway were used to verified the proposed method. The contribution of the study is of great practical significance. However, there are some defects:

1. In the introduction, the citation style is not standard. For example, “Yosuke[3]” should be written as “Tsubokawa et al. [3]”. “Q Chen et al. [5]” should be “Chen et al. [5]”. Please review other citations to ensure they are correctly formatted.

2. Line 172, in my opinion, the symbol of the matrix F should be bold and should not be italicized.

3. In section 3, how is the parameters of the CNN-Bi-LSTM in Table 2 (line 200). be determined?

4. There are some misleading naming of figures and tables in the manuscript. For example, line 188, “Figure 2” should be “Figure 1”. Line 195 “Figure 3” and line 199 “Figure 1” should both be “Figure 2”. Line 200 “Table 2” should be “Table 1”. And lines 339 and 341 are repeated. Please review other figures and tables to ensure they are correctly named.

Comments on the Quality of English Language

The Quality of English Language can be improved.

Author Response

Dear reviewer,

Thank you very much for taking the time to review this manuscript. Please find the detailed responses below and the corresponding revisions highlighted. 

Comments 1: In the introduction, the citation style is not standard. For example, “Yosuke[3]” should be written as “Tsubokawa et al. [3]”. “Q Chen et al. [5]” should be “Chen et al. [5]”. Please review other citations to ensure they are correctly formatted.

Response 1: Thank you very much for your feedback. We have checked and revised all citation writing in the manuscript.

Comments 2: Line 172, in my opinion, the symbol of the matrix F should be bold and should not be italicized.

Response 2: Your opinion is very correct. In the new version of the manuscript, a new method has been adopted for the derivation of formulas in this section, with vectors and matrices represented in bold.

Comments 3:In section 3, how is the parameters of the CNN-Bi-LSTM in Table 2 (line 200). be determined?

Response 3:Dear reviewer, the sampling interval for the real detection data used in this article is 0.25 meters. The input sequence length for this article is 500 data points, with a range of approximately 125 meters. At present, the maximum wavelength range for evaluating track irregularities is 120 meters, so 500 data points were used as inputs. Predicting track irregularities through vehicle acceleration can mainly predict large fluctuation trends, but it is difficult to describe specific details. Therefore, a 1x10 convolution kernel was used in 1D-CNN, which can cover irregularities with wavelengths of over 2 meters. The selection of the number of hidden layer nodes in the BI-LSTM layer and fully connected layer is based on some comparative experiments, and it is necessary to avoid overfitting caused by excessive number of nodes.

Comments 4:There are some misleading naming of figures and tables in the manuscript. For example, line 188, “Figure 2” should be “Figure 1”. Line 195 “Figure 3” and line 199 “Figure 1” should both be “Figure 2”. Line 200 “Table 2” should be “Table 1”. And lines 339 and 341 are repeated. Please review other figures and tables to ensure they are correctly named.

Response 4:Thank you for pointing out the error. I have checked and revised the names and numbers of all the figures and tables in the manuscript.

Reviewer 3 Report

Comments and Suggestions for Authors

This paper presents a track irregularity identification method based on CNN-Bi-LSTM, which effectively extracts features from vehicle acceleration data and enables point-by-point prediction of irregularities. The introduction of the exponential weighted mean square error with priority inner fitting (EIF-MSE) as a loss function is a noteworthy innovation, enhancing the accuracy of big value data prediction and reducing the risk of false alarms. Overall, this paper holds high academic value and practical significance, and I recommend its acceptance after addressing the following revision suggestions:

Point 1: It is suggested to provide an overview of the latest research progress in the field of track irregularity identification in the introduction section. This will help readers understand the context and significance of your work in relation to existing literature.

Point 2: Please provide detailed information about the specific method employed for collecting vehicle acceleration data. This will enhance the reproducibility of your research and allow readers to better understand the data characteristics.

Point 3: In the second section, where you discuss the model's input data segment length of 500, it would be beneficial to explain the rationale behind choosing this specific length. Clarifying the reasons for selecting this parameter will enhance the readers' understanding of the model's input requirements.

Point 4: Please provide a detailed explanation of the basis for determining the specific parameters used in the model.

Point 5: I recommend conducting a thorough proofreading of the paper to identify and correct any spelling mistakes to ensure the paper's readability.

Point 6: There are some numbering discrepancies in the figures and tables. For instance, there are two Figure 1s but no Figure 2, and there are two Table 2s but no Table 1. Please carefully review and correct these errors to ensure clarity and consistency throughout the paper.

Point 7: As mentioned in the introduction, track irregularity data play a critical role in infrastructural health management and track maintenance. To provide readers with a broader context, I suggest considering the inclusion of the following relevant literature in your introduction section, if appropriate:

Correlation Analysis between Rail Track Geometry and Car-Body Vibration Based on Fractal Theory. https://doi.org/10.3390/fractalfract6120727.

Author Response

Dear reviewer,

Thank you very much for taking the time to review this manuscript. Please find the detailed responses below and the corresponding revisions highlighted.

Comments 1: It is suggested to provide an overview of the latest research progress in the field of track irregularity identification in the introduction section. This will help readers understand the context and significance of your work in relation to existing literature..

Response 1: Thank you for your suggestion. In the first section, we have added an introduction to the latest research results on identifying and maintaining track irregularities.

Comments 2: Please provide detailed information about the specific method employed for collecting vehicle acceleration data. This will enhance the reproducibility of your research and allow readers to better understand the data characteristics.

Response 2: Thank you for your suggestion. We have added an introduction to data collection methods in Section 5.2.

Comments 3:In the second section, where you discuss the model's input data segment length of 500, it would be beneficial to explain the rationale behind choosing this specific length. Clarifying the reasons for selecting this parameter will enhance the readers' understanding of the model's input requirements.

Response 3:Dear reviewer, the sampling interval for the real detection data used in this article is 0.25 meters. The input sequence length for this article is 500 data points, with a range of approximately 125 meters. At present, the maximum wavelength range for evaluating track irregularities is 120 meters, so 500 data points were used as inputs

Comments 4:Please provide a detailed explanation of the basis for determining the specific parameters used in the model.

Response 4:Predicting track irregularities through car body acceleration can mainly predict large fluctuation trends, but it is difficult to describe specific details. Therefore, a 1x10 convolution kernel was used in 1D-CNN, which can cover irregularities with wavelengths of over 2 meters. The selection of the number of hidden layer nodes in the BI-LSTM layer and fully connected layer is based on some comparative experiments, and it is necessary to avoid overfitting caused by excessive number of nodes.

Comments 5:I recommend conducting a thorough proofreading of the paper to identify and correct any spelling mistakes to ensure the paper's readability.

Response 5:Thank you for your advice. We have carefully checked the spelling and grammarmistakes in the paper, and also invited English teacher to check the paper.

Comments 6:There are some numbering discrepancies in the figures and tables. For instance, there are two Figure 1s but no Figure 2, and there are two Table 2s but no Table 1. Please carefully review and correct these errors to ensure clarity and consistency throughout the paper.

Response 6:Thank you for pointing out the error. I have checked and revised the names and numbers of all the figures and tables in the manuscript.

Comments 7:As mentioned in the introduction, track irregularity data play a critical role in infrastructural health management and track maintenance. To provide readers with a broader context, I suggest considering the inclusion of the following relevant literature in your introduction section, if appropriate:

Correlation Analysis between Rail Track Geometry and Car-Body Vibration Based on Fractal Theory. https://doi.org/10.3390/fractalfract6120727.

Response 7:Thank you for your suggestion. The literature you mentioned has important reference value for this article and has been added to the first section

Reviewer 4 Report

Comments and Suggestions for Authors

The article submitted for review, "A method for identifying high-speed railway track irregularities based on CNN-Bi-LSTM," is devoted to the development of methods for assessing railway track irregularities. The authors rely on the modern neural network paradigm of modeling complex systems and processes. In general, the article is interesting for a wide range of specialists in this field of scientific knowledge, but it is not without drawbacks.

1) There are many abbreviations and abbreviations in the article, the description of which is either not given or partially given. To improve the quality of work, it is necessary to prepare a separate sheet of abbreviations and designations.

2) In expressions (1), (2), (3), The authors introduce a discrete variable t, which denotes time. However, at t>1, we are talking about the future tense, which is difficult to imagine. The authors need to describe this variable, as well as explain the correctness of the choice of discretionary numbers.

3) In lines 167-168, the value 30 is assigned to the beta variable, which the authors do not explain in any way.

4) In the diagram shown in Figure 1, the authors introduce two time series, one is designated as acceleration, the second as displacement. From this figure and further explanation, it is not clear whether these quantities are related to each other, that is, the displacements obtained by double integration from acceleration, or these are two separately measured sequences. In general, this issue requires additional clarification.

5) In the article, the authors constantly mention the car body, but the article is devoted to the problem of assessing the irregularities of the railway track. Apparently there is some linguistic interpretation of the railway trolley.

6) In terms of verifying the simulation results, the authors rely on the results of experiments conducted by the Inspection of Railways of the People's Republic of China. The authors do not describe the conditions of the experiment or the experimental equipment. It is impossible to make an assessment of the correspondence of experimental and forecast values from the above material.

7) In conclusion, the authors summarize the work on the assessment of vertical vibrations of the car body!

Author Response

Dear reviewer,

Thank you very much for taking the time to review this manuscript. Please find the detailed responses below and the corresponding revisions highlighted. 

Comments 1: There are many abbreviations and abbreviations in the article, the description of which is either not given or partially given. To improve the quality of work, it is necessary to prepare a separate sheet of abbreviations and designations.

Response 1: Thank you for your suggestion. We have added a list of abbreviations at the end of the article.

Comments 2: In expressions (1), (2), (3), The authors introduce a discrete variable t, which denotes time. However, at t>1, we are talking about the future tense, which is difficult to imagine. The authors need to describe this variable, as well as explain the correctness of the choice of discretionary numbers.

Response 2: Dear reviewer, the track irregularity prediction algorithm proposed in this article is not a real-time prediction model. The input of the model is historical detection data of vehicle acceleration with a length of 500 points. The data is sampled at equal intervals, with a sampling interval of 0.25 meters. When t=1, it represents the first point in the sequence, and when t=2, it represents the second point. The derivation process of this section has been modified in the new version of the manuscript to make it easier for readers to understand.

Comments 3:In lines 167-168, the value 30 is assigned to the beta variable, which the authors do not explain in any way.

Response 3:Thank you for your kind comments. According to Ref. [30], the determinationof β mainly depends on the detection frequency of monitoring data. The higher the detection frequency is, the greater the value of β is. In this paper, the effect of different values of βis compared for many times, when β is taken as 30, the trend component extracted is the best for the subsequent residual life prediction. When the monitoring frequency changes, the value of β needs to be adjusted.

Comments 4:In the diagram shown in Figure 1, the authors introduce two time series, one is designated as acceleration, the second as displacement. From this figure and further explanation, it is not clear whether these quantities are related to each other, that is, the displacements obtained by double integration from acceleration, or these are two separately measured sequences. In general, this issue requires additional clarification.

Response 4:Dear reviewer, thank you for your questions. The track irregularity detection data and vehicle acceleration detection data used in this article are collected by two independent systems. By integrating the acceleration of the axle box twice, it is indeed possible to obtain.We have added an introduction to data collection methods in Section 5.2.

During the data collection process, the track irregularity detection system and the vehicle acceleration detection system are deployed on the same carriage, ensuring the synchronization of data collection. The track irregularity detection system uses the inertial reference method to measure through gyroscopes and displacement sensors. The collection of vehicle acceleration is carried out using an acceleration sensor installed on the bottom plate of the vehicle, which is located on the same cross-section as the track irregularity detection system. The track irregularity and vehicle acceleration are both sampled at equal intervals, with a sampling interval of 0.25 meters.

It is indeed possible to obtain track irregularities by integrating the acceleration of the axle box twice, but many literature points out that this method is susceptible to environmental interference and has limited measurement accuracy. The track irregularity detection data in this article was collected by professional detection vehicles based on the inertial reference method as the prediction target of the model. The purpose of this article is to use more easily collected vehicle acceleration data to predict track irregularities, thereby improving detection frequency at a lower cost.

Comments 5:In the article, the authors constantly mention the car body, but the article is devoted to the problem of assessing the irregularities of the railway track. Apparently there is some linguistic interpretation of the railway trolley.

Response 5:Thank you for your question,We have added an introduction to data collection methods in Section 5.2. The collection of vehicle acceleration is carried out using an acceleration sensor installed on the bottom plate of the vehicle, which is located on the same cross-section as the track irregularity detection system.

Track irregularity refers to the deviation of the geometric shape, size, and spatial position of the track from its normal state. Currently, it is mainly detected by professional inspection vehicles on a regular basis, and the detection cost is relatively high. Many literature points out that there is a high correlation between vehicle acceleration and track roughness. The method proposed in this article aims to predict track irregularities through vehicle acceleration. Because the detection of vehicle acceleration is very convenient, it can be collected through daily passenger trains, thereby increasing the detection frequency and reducing detection costs.

Comments 6:In terms of verifying the simulation results, the authors rely on the results of experiments conducted by the Inspection of Railways of the People's Republic of China. The authors do not describe the conditions of the experiment or the experimental equipment. It is impossible to make an assessment of the correspondence of experimental and forecast values from the above material.

Response 6:Thank you for your suggestion. We have added explanations on the data collection process, collection conditions, and equipment installation used in the article in Section 5.2

Comments 7:In conclusion, the authors summarize the work on the assessment of vertical vibrations of the car body!

Response 7:What you said is very correct. This article uses vehicle acceleration data to predict track irregularities. In the conclusion section, the model structure and the proposed EIF-MSE loss function are summarized, and the effectiveness of the proposed method is demonstrated.

Round 2

Reviewer 1 Report

Comments and Suggestions for Authors

this version is OK, I have not any comments

Reviewer 4 Report

Comments and Suggestions for Authors

The authors answered all the questions in detail and made a sufficient number of edits to the article.